# A Novel Biomarker for Acute Kidney Injury, Vanin-1, for Obstructive Nephropathy: A Prospective Cohort Pilot Study

**DOI:** 10.3390/ijms20040899

**Published:** 2019-02-19

**Authors:** Satoshi Washino, Keiko Hosohata, Masashi Oshima, Tomohisa Okochi, Tsuzumi Konishi, Yuhki Nakamura, Kimitoshi Saito, Tomoaki Miyagawa

**Affiliations:** 1Department of Urology, Jichi Medical University Saitama Medical Center, 1-847, Amanuma-cho, Omiya-ku, Saitama 330-8503, Japan; m_ohshima_amihsho_m@yahoo.co.jp (M.O.); tsuzumi0203@gmail.com (T.K.); yuhki.dreamskyward@icloud.com (Y.N.); lespaul991200@gmail.com (K.S.); sh2-miya@jichi.ac.jp (T.M.); 2Education and Research Center for Clinical Pharmacy, Osaka University of Pharmaceutical Sciences, 4-20-1 Nasahara, Takatsuki 569-1094, Japan; hosohata@gly.oups.ac.jp; 3Department of Radiology, Jichi Medical University Saitama Medical Center, Saitama 330-8503, Japan; t_shachi@dj8.so-net.ne.jp

**Keywords:** vanin-1, NGAL, KIM-1, NAG, hydronephrosis, urinary biomarker

## Abstract

**Background**: Vanin-1 is a novel acute kidney injury (AKI) biomarker that has not been clinically investigated as a biomarker for obstructive nephropathy. This study investigated the diagnostic value of vanin-1 as a biomarker for adult obstructive nephropathy by comparing it to existing AKI biomarkers. **Methods**: A total of 49 patients, 21 controls, and 28 hydronephrosis (HN) cases were assessed. AKI biomarkers in bladder (BL) urine and renal pelvic (RP) urine in the HN group were compared to each BL marker in the control group. In a subgroup of cases receiving interventions for obstructive nephropathy, the BL values of each biomarker were assessed after the intervention. **Results**: RP vanin-1 levels were significantly higher while BL vanin-1 levels were marginally higher in the HN group than in the control group. The area under the receiver operating characteristics curve values for RP and BL vanin-1 were 0.9778 and 0.6386, respectively. In multivariate analyses, BL vanin-1 and N-acetyl-β-D-glucosaminidase (NAG), but not kidney injury molecule-1 (KIM-1) or neutrophil gelatinase-associated lipocalin (NGAL), were independent factors for predicting the presence of HN. In cases receiving interventions, vanin-1 decreased significantly from 1 week after the intervention in cases of moderate to severe obstructive nephropathy compared to RP values at baseline. **Conclusion**: Urinary vanin-1 is a useful biomarker to detect and monitor the clinical course of obstructive nephropathy.

## 1. Introduction

Obstructive nephropathy is a common cause of acute kidney injury (AKI), chronic kidney disease (CKD), and end-stage renal disease [1,2]. Obstruction of urinary flow increases intra-tubular pressure and can lead to impaired renal blood flow and inflammatory processes.

Plasma or urine markers, including neutrophil gelatinase-associated lipocalin (NGAL), kidney injury molecule-1 (KIM-1), and N-acetyl-β-D-glucosaminidase (NAG), are promising potential biomarkers for the early detection and monitoring of AKI and CKD [3,4]. NGAL, KIM-1, and NAG have also been studied as biomarkers for obstructive nephropathies, such as ureteropelvic junction obstruction (UPJO) and ureteral calculi and are useful early biomarkers of progressive renal damage. Therefore, they could have a potential role in predicting long-term renal outcomes in obstructive nephropathy and are more sensitive than changes in the serum levels of creatinine [5,6,7,8]. However, the clinical usefulness of these biomarkers is limited by a lack of sensitivity or specificity [9,10,11]; therefore, a new AKI biomarker for obstructive nephropathy is needed.

Recently, vanin-1 was identified as another AKI biomarker, and urinary vanin-1 has been shown to have superior predictive value for cisplatin-induced AKI than KIM-1, NGAL, or NAG [12,13,14]. Vanin-1 is highly and selectively expressed in renal tubular epithelial cells [15]. Therefore, tubular injury caused by increased intra-tubular pressure might cause vanin-1 to be detectable in the urine. This led us to speculate that vanin-1 could be a marker of obstructive nephropathy. Vanin-1 is released from renal tubular cells in a rat ureteral obstructive model [16]. However, the clinical usefulness of vanin-1 for obstructive nephropathy has not been investigated.

We investigated the diagnostic value of vanin-1 as a urinary biomarker for adult obstructive nephropathy by comparing it to existing AKI biomarkers, such as KIM-1, NGAL, and NAG.

## 2. Results

### 2.1. Patient Characteristics

Of the 54 patients enrolled (23 in the control group and 31 in the hydronephrosis (HN) group), 2 were excluded from the control group (1 had muscle-invasive bladder cancer and the other had prostate cancer invading the seminal vesicle) and 3 were excluded from the HN group (2 did not undergo an Tc-99m mercaptoacetyltriglycine (MAG3) renal scan and 1 failed to have urine samples collected at the first assessment). A total of 49 patients, 21 controls and 28 HN cases (16 extrinsic upper urinary tract obstruction (UUTO) and 12 intrinsic UUTO), were analyzed. Of the 16 cases with extrinsic UUTO, 14 were malignant ureteral obstruction (MUO) and 2 were ureteral obstruction caused by endometriosis or abdominal artery aneurysm; 10 required indwelling double J (DJ) stents to improve the obstruction during the study period. Of the 12 cases with intrinsic UUTO, 10 were UPJO and 2 were ureteral stricture; 5 required indwelling DJ stents; and 2 cases underwent surgery during the study. One patient, presented below in the case presentation section, was initially diagnosed with right HN by MUO, but improved after intervention; she was then diagnosed with left HN by MUO. This patient was counted as two cases and all other cases are represented as individual patients. The reasons for a referral to our department in the control group included prostate cancer (43%), bladder tumors (38%), and others (19%).

The characteristics of the control and HN cases are shown in Table 1. The mean age in the HN group was 53.85 ± 14.55 years, significantly lower than in the control group (67.5 ± 10.73; *p* = 0.0004). There was a significantly higher percentage of women in the HN group than in the control group (*p* = 0.0008). Overall, 14 (50%) right and 14 (50%) left kidneys were affected in the HN group. The control group included 3 cases (14%) of diabetic and hypertensive nephropathy, while the HN group included 2 cases (7%) of drug-induced nephropathy.

### 2.2. UUTO Patterns According to Tc-99m Mercaptoacetyltriglycine(MAG3) Renal Scan Results

There were no cases with pattern 0 scan results (normal renal scan pattern). In both extrinsic and intrinsic UUTO, the normalized effective renal plasma flow-ortho-iodohippurate (ERPF-OIH) and split in the MAG3 renal scan and the estimated glomerular filtration rate (eGFR) generally decreased with an increasing UUTO pattern (Figure 1A,B). The UUTO pattern was inversely correlated with normalized ERPF-OIH (*r* = −0.8654, *p* < 0.0001) and eGFR (*r* = −0.4581, *p* = 0.0089). Normalized ERPF-OIH in the affected kidney had a strong correlation with eGFR (*r* = 0.7482, *p* < 0.0001). Both normalized ERPF-OIH and eGFR were higher in intrinsic UUTO than in extrinsic UUTO (100.8 ± 58.57 vs. 76.76 ± 40.83 mL/min, *p* = 0.2358 in ERPF-OIH; 73.13 ± 20.36 vs. 55.78 ± 17.11 min/mL/1.73 m^2^, *p* = 0.0316 in eGFR).

### 2.3. Urinary Biomarkers

Serum levels of creatinine did not differ between the HN and control groups (Figure 2A). However, eGFR was significantly lower in extrinsic UUTO (but not in intrinsic UUTO or all UUTO cases) than in controls (55.78 ± 17.11, 73.13 ± 20.36, and 62.84 ± 20.09 vs. 68.8 ± 13.98 mL/min/1.73 m^2^, respectively; *p* = 0.0185 for extrinsic UUTO vs. control; Figure 2B).

Levels of bladder vanin-1 were marginally higher, and NGAL and NAG were significantly higher in all UUTO cases than in controls, whereas KIM-1 did not differ between the two groups (vanin-1: 5.088 ± 8.135 vs. 0.6462 ± 1.117 ng/mgCre, *p* = 0.0927; KIM-1: 2.335 ± 4.697 vs. 0.8505 ± 0.9732 ng/mgCre, *p* = 0.7179; NGAL: 84.22 ± 166.8 vs. 8.035 ± 17.25 ng/mgCre, *p* = 0.0449; and NAG: 7.229 ± 5.791 vs. 2.902 ± 1.547 U/mgCre, *p* = 0.0070) (Figure 2C–F and Appendix A). Renal pelvic (RP) vanin-1 (16.43 ± 22.92 ng/mgCre, *p* < 0.0001), KIM-1 (18.3 ± 35.5, *p* = 0.0003), NGAL (446 ± 1020, *p* = 0.0078), and NAG (15.39 ± 14.80 U/mgCre, *p* < 0.0001) were significantly higher in the HN group than in controls.

Levels of bladder vanin-1 (6.727 ± 9.862 mg/mgCre, *p* = 0.0895) were marginally higher, and NGAL (117.7 ± 205.1 ng/mgCre, *p* = 0.0073) and NAG (8.259 ± 5.396 U/mgCre, *p* = 0.0017), but not KIM-1, were significantly higher in extrinsic UUTO than in controls. Levels of all AKI biomarkers in RP urine were significantly higher in extrinsic UUTO than in the controls. The levels of each urinary AKI biomarker were lower in intrinsic UUTO than in extrinsic UUTO cases (Figure 2C–F and Appendix A). RP vanin-1 levels were significantly higher in intrinsic UUTO cases than in controls (*p* = 0.0044), but the levels of other biomarkers did not differ significantly compared to controls.

### 2.4. Multivariate Analyses to Identify Factors that Predict HN

Next, we assessed whether the four urinary AKI biomarkers or eGFR were independent markers for the prediction of the presence of HN. Univariate and multivariate analyses were performed by comparing the levels of each bladder biomarker in the HN and control groups. In the univariate analyses, vanin-1 and NAG, but not KIM-1 or eGFR, were significantly associated with the presence of HN. NGAL was marginally associated with the presence of HN. In the multivariate analyses, vanin-1 (HR 1.546, 95% confidence interval (CI) 1.046–2.283, *p* = 0.029) and NAG (HR 1.458, 95% CI 1.067–1.992, *p* = 0.018) were independent predictors (Table 2).

### 2.5. Predictive Ability of Each Biomarker

The area under the curve (AUC) of the receiver operating characteristics (ROC) curve analysis for vanin-1, KIM-1, NGAL, and NAG was 0.9778, 0.8825, 0.7619, and 0.9143, respectively, in RP urine and 0.6386, 0.5731, 0.6990, and 0.7423 in bladder (BL) urine (Figure 3). To determine the cut-off for each marker, the levels of each biomarker in RP urine and BL urine in HN cases were compared to those in BL urine in the control group. The cut-offs in the RP/BL urine were set at 3.315/3.315 ng/mgCre, 2.240/2.190 ng/mgCre, 13.05/13.99 ng/mgCre, and 5.755/5.410 U/mgCre for vanin-1, KIM-1, NGAL, and NAG, respectively. The sensitivity and specificity of each biomarker in RP urine was 0.867 and 0.952, 0.667 and 0.952, 0.600 and 0.952, and 0.733 and 0.952 for vanin-1, KIM-1, NGAL, and NAG, respectively, whereas those in bladder urine were 0.393 and 0.952, 0.214 and 0.952, 0.543 and 0.952, and 0.536 and 0.952. KIM-1 was less useful as a biomarker for UUTO than vanin-1 because the AUCs of KIM-1 in RP and bladder urine were less than those of vanin-1, respectively (Figure 3). Furthermore KIM-1 was not a predictive factor for the presence of HN in either the univariate or multivariate analyses (Table 2); therefore, we omitted KIM-1 from subsequent analyses.

### 2.6. Urinary Biomarker Levels According to the UUTO Pattern

Vanin-1, NGAL, and NAG levels in RP urine generally increased with an increasing UUTO pattern in extrinsic UUTO cases, indicating that more kidney injury occurred in patients with more severe UUTO (Figure 4A). Bladder vanin-1 levels in pattern 3 extrinsic UUTO cases were higher than those in pattern 1 or 2 cases (14.90 ± 13.55 ng/mgCre in pattern 3 vs. 7.160 ± 6.850 or 2.905 ± 4.328 ng/mgCre in pattern 1 and 2, respectively), whereas levels in pattern 4 were undetectable, despite very high levels in RP urine (Figure 4A). The pattern of NGAL levels was similar to that of vanin-1 (Figure 4A). By contrast, bladder NAG levels were detectable in pattern 4 cases and were higher than those in pattern 1–3 cases (9.378 ± 4.460 in pattern 4 vs. 8.56 ± 11.35, 6.585 ± 3.914, and 8.524 ± 3.761 U/mgCre in patterns 1, 2, and 3, respectively) (Figure 4A). In intrinsic UUTO, bladder vanin-1 levels in pattern 4 cases were lower than the cut-off. By contrast, bladder NAG levels generally increased with an increasing UUTO pattern (Figure 4B), consistent with extrinsic UUTO cases.

### 2.7. Combination of Bladder Vanin-1 and NAG

The sensitivity of vanin-1 was lower in bladder urine than in RP urine (Figure 3C), which was mainly due to it being below the cut-off in pattern 4 UUTO cases (Figure 4). When the levels of bladder vanin-1 or NAG above the cut-off were considered positive, it yielded a better sensitivity (0.6786) and an acceptable specificity (0.9048) compared to vanin-1 or NAG as a single marker.

### 2.8. Monitoring Kidney Injury after Intervention

After intervention, normalized ERPF-OIH improved in both pattern 1–2 UUTO and pattern 3—4 UUTO cases; however, the improvements were not statistically significant (Figure 5A). eGFR increased significantly 1 week and 4 weeks after intervention in pattern 3–4 UUTO cases, but the increase in pattern 1–2 UUTO cases was not statistically significant (Figure 5B). When the levels of each AKI biomarker were compared in bladder urine before and after intervention, they did not decrease significantly after intervention (Figure 5C). When assessed in individual cases, the levels of AKI biomarkers decreased by 1 week after intervention in most cases with higher levels at baseline. In those with lower levels at baseline, the levels increased at 1 week and then decreased at 4 weeks (Appendix A). When comparing the RP markers before intervention with the bladder values after intervention, vanin-1 levels decreased significantly at 1 week and 4 weeks in pattern 3–4 UUTO cases (21.97 ± 26.69 (baseline) to 5.424 ± 6.572 and 3.760 ± 4.691 ng/mgCre at 1 week (*p* < 0.05) and 4 weeks (*p* < 0.05), respectively). NAG levels were significantly lower at 4 weeks (19.16 ± 16.73 (baseline) vs. 11.84 ± 11.92 and 7.780 ± 5.421 U/mgCre at 1 week (not significant) and 4 weeks (*p* < 0.05), significantly) (Figure 5D). By contrast, NGAL levels did not decrease significantly (Figure 5D).

### 2.9. Case Presentation of a Patient with Asynchronous Bilateral Extrinsic UUTOs

A female patient was referred to our department with right HN by MUO, suffering from uterine cervical cancer with pelvic lymph node involvement. Her right HN was categorized as pattern 3 UUTO (split, 38.2%; ERPF-OIH, 91.3 mL/min). Bladder vanin-1 and NAG levels were 5.43 ng/mgCre and 4.14 U/mgCre, respectively. The DJ stent had been indwelled into her right upper urinary tract (UUT) when the vanin-1 and NAG levels in her right RP urine were 10.86 ng/mgCre and 4.95 U/mgCre. After 1 week, the bladder vanin-1 and NAG levels decreased to 0.00 ng/mgCre and 1.83 U/mgCre, respectively (Figure 6). An MAG3 renal scan at 4 weeks showed improved MAG3 clearance in the right UUT (green line in the MAG3 renal scan graph), although the split (38.0%) and ERPF-OIH (92.3 mL/min) were comparable; some delayed clearance at her left UUT was observed (pattern 1 UUTO, red line in the MAG3 renal scan graph). At 9 weeks after the initial intervention, a follow-up computed tomography revealed HN in her left UUT and the MAG3 renal scan indicated a pattern 3 UUTO (split, 29.4%; ERPF-OIH, 56.1 mL/min). The bladder vanin-1 and NAG levels increased again to 38.4 ng/mgCre and 13.28 U/mgCre, respectively. A DJ stent was indwelled into her left UUT (RP vanin-1, 39.1 ng/mgCre; RP NAG, 31.9 U/mgCre). One-week after the second intervention, her bladder vanin-1 and NGAL levels decreased to 17.87 ng/mgCre and 12.58 U/mgCre, respectively, and further decreased to 12.87 ng/mgCre and 11.30 U/mgCre at 4 weeks. A renal scan performed at 4 weeks revealed that the left UUTO had improved (split, 43.3%; ERPF-OIH, 96.9 mL/min).

## 3. Discussion

We performed a pilot study to evaluate the diagnostic values of the urinary kidney injury marker, vanin-1, as a biomarker for UUTO by comparing it with the existing urinary AKI biomarkers, KIM-1, NGAL, and NAG. Vanin-1 was highly sensitive and specific in RP urine but, had lower sensitivity in bladder urine. It was also useful for monitoring the clinical course after treatment for UUTO. NGAL and NAG were also useful biomarkers for UUTO-induced kidney injury, but the usefulness of KIM-1 was limited.

Vanin-1 is an epithelial ectoenzyme that shows pantetheinase activity; it catalyzes the conversion of pantetheine into pantothenic acid and cysteamine [15,17]. The highest levels of vanin-1 mRNAs are found in the kidney, where tubular epithelial cells selectively express vanin-1 transcripts, although vanin-1 transcripts are expressed ubiquitously in mouse organs [15]. Recently, we and others have demonstrated that vanin-1 is an AKI biomarker with good diagnostic value in various kidney diseases, such as diabetic nephropathy and drug-induced renal tubular injury [12,13,14,18]. Obstruction of urinary flow increases intra-tubular pressure, resulting in renal tubular injury. Therefore, renal tubular injury caused by increased intra-tubular pressure could cause vanin-1 to leak into the urine of patients with UUTO. Hosohata et al. demonstrated that vanin-1 was released from renal tubular cells in a rat ureteral obstructive model [16].

In the present study, we demonstrated that RP urinary vanin-1 was highly sensitive and specific and that the levels in RP urine correlated with the severity of obstruction in extrinsic UUTO (Figure 3A and Figure 4A). Bladder vanin-1 was less sensitive than RP vanin-1, which could partially be explained by the fact that bladder vanin-1 levels were below the cut-off in pattern 4 UUTO (Figure 4A,B). It would be reasonable to consider that, in pattern 4 UUTO, the urinary obstruction was complete and so there was no urine flow from the affected kidney to the bladder. In addition, vanin-1 might also not be absorbed from the affected kidney into the bloodstream or excreted to the urine from the contralateral kidney. In a study that used a rat UUTO model, serum levels of vanin-1 in UUTO model rats were not different from those in sham rats, despite quite different urinary vanin-1 levels in UUTO (high levels) and sham rats (low) [16]. These vanin-1 characteristics might be useful to identify severe UUTO cases (cases with high RP vanin-1 levels with low bladder vanin-1 levels) and these patients should immediately undergo intervention. The downside of these characteristics is that it is difficult to know whether very low or undetectable bladder vanin-1 levels means no, very mild, or complete UUTO. Thus, an MAG3 renal scan or RP values should be checked. However, only a few institutes can conduct MAG3 renal scans, which are expensive and expose the patient to radiation. Collecting RP urine is an invasive examination. We believe this downside of bladder vanin-1 could be overcome by checking other urinary markers, such as NAG, or other serum AKI biomarkers in combination with bladder vanin-1.

Intrinsic UUTO, such as UPJO, and ureteral stricture cases had lower AKI biomarker levels and higher ERPF-OIH and eGFR compared to extrinsic UUTO cases (Figure 1 and Figure 2C–F). The process of adult UPJO or ureteral stricture is usually chronic, which may be associated with less kidney injury. However, even with such mild kidney injury, RP vanin-1 levels, but no other AKI biomarkers, detected the presence of kidney injury (Figure 2C and Figure 4B).

Vanin-1 was also useful for monitoring the clinical course of UUTO after intervention. Vanin-1 levels decreased starting from 1 week after intervention in moderate to severe UUTO cases compared to RP values before intervention (Figure 5D). When we examined how bladder vanin-1 levels changed in individual cases, ~50% of cases first experienced an increase in levels after intervention followed by a decrease (Appendix A). This suggests that urine flow had improved in the affected kidney after intervention, which led to increased bladder vanin-1 values, and then kidney injury improved following re-canalization of the UUTO, which resulted in decreased levels later. We also described a case report of a patient with extrinsic UUTO who experienced a decrease and increase in vanin-1 corresponding to the improvement and worsening of UUTO (Figure 6).

Taken together, these results suggest that vanin-1 is useful for assessing the presence and severity of UUTO, and monitoring the clinical course. However, a reasonable vanin-1 cut-off after intervention has yet to be determined because most bladder vanin-1 levels at 4 weeks were above the screening cut-off of 3.32 ng/mgCre in extrinsic UUTO (Appendix A).

When comparing the predictive ability of four urinary markers for UUTO, bladder vanin-1 and NAG were independent factors for predicting the presence of HN in multivariate analyses (Table 2). NGAL was a predictive factor in univariate analyses, but not multivariate analyses (Figure 2E and Table 2). Bladder KIM-1 was less useful as a biomarker for UUTO. Urbschat et al. showed that urinary NGAL levels were significantly higher in ureteral calculi-induced obstructive nephropathy than in healthy controls whereas urinary KIM-1 levels were not [11]; our results are consistent with those findings. NAG has some different characteristics compared to vanin-1 or NGAL. The levels of bladder NAG were higher in pattern 4 UUTO, which were probably cases with complete obstruction, than pattern 1–3 UUTO cases in extrinsic UUTO (Figure 4A). NAG might be absorbed from the affected kidney into the bloodstream and excreted to the urine from the contralateral kidney, although there has been no evidence that demonstrates this speculation. These characteristics of NAG might compensate for the downside of vanin-1, because bladder vanin-1 levels are undetectable in complete UUTO. We demonstrated that the combination of bladder vanin-1 and NAG a exhibited higher sensitivity and acceptable specificity compared to each marker alone. However, it will be necessary to identify serum AKI biomarkers to distinguish very mild UUTO from complete UUTO. MicroRNAs (miRNAs) have emerged as novel biomarkers for AKI [19,20]. miR-10a is a renal tubule-specific miRNA that decreases in the urine and kidney tissues in rodent models of nephropathy [20]. In humans, decreased plasma levels of miR-10a predict AKI in critical patients [21]. miR-29a is highly expressed in the kidney, and low serum levels of miR-29a predict AKI in intensive care unit patients, and are correlated with AKI severity [21]. However, miRNAs have been not been studied in UUTO-induced kidney injury.

This study had some limitations. First, the number of cases was relatively small, which did not allow us to draw unequivocal conclusions. Second, the comparison between the HN and control groups was performed in two different age and sex populations. However, there was no statistically significant correlation between age and the values of each urinary AKI biomarker in the control group (vanin-1, *p* = 0.1842; KIM1, *p* = 0.1053; NGAL, *p* = 0.3116; and NAG, *p* = 0.0801 in Pearson’s correlation coefficient). Third, although diuretic Tc-99mMAG3 renal scans are usually used to evaluate UUTO to distinguish obstructive HN from non-obstructive HN, we did not use diuretics in this study because the use and timing of diuretics depends on the obstruction status, which would affect the results of normalized ERPF-OIH and make comparing values at different time points difficult. Fourth, there is no widely accepted categorization of UUTO. However, our categorization had a good correlation with both normalized ERPF-OIH and eGFR. Fifth, other kidney diseases might affect the levels of AKI biomarkers.

## 4. Patients and Methods

This prospective observational study was approved on 14 July 2015 by the Ethics Committee of Jichi Medical University (RinS16–003) and conducted between October 2015 and September 2017 in accordance with the Declaration of Helsinki. All subjects gave their informed consent for inclusion before they participated in the study.

Among the patients who visited our department, adults who had unilateral HN were recruited to the HN group and those without HN were recruited to the control group. Patients with bilateral HN, single upper urinary tract, urolithiasis, or symptomatic urinary infection were excluded. The presence of HN was assessed by computed tomography within 3 months of enrollment. In total, 54 patients (31 in the HN group and 23 in the control group) participated after providing written informed consent. In some analyses, HN cases were divided into two groups: Extrinsic UUTO, represented by MUO, and intrinsic UUTO, represented by UPJO because extrinsic UUTO is an acute process whereas intrinsic UUTO is a chronic process.

In the control group, patients with a disease that could influence urine flow, such as muscle invasive bladder cancer or prostate cancer invading the seminal vesicles or bladder, were excluded from the study. Patients who failed to undergo planned radiological tests or for whom samples were not collected at the first assessment were also excluded.

The degree of UUTO in the HN group was assessed using Tc-99m MAG3 renal scans (E.Cam imaging system, Canon Medical System, Otawara, Japan). Tc-99mMAG3 was injected intravenously (IV) and the uptake and clearance of Tc-99mMAG3 in both kidneys was tracked for 30 min using a gamma camera (Canon Medical System, Otawara, Japan). Tc-99mMAG3 clearance is highly correlated with ERPF [22]. Split renal function was calculated by dividing the radioactive tracer accumulation from each side in the first 2 min by the total accumulation in both kidneys over the same period. Normalized ERPF-OIH was calculated using Itoh’s method [23]. HN cases were categorized into five patterns according to the following criteria: Pattern 0 (normal: No delayed peak (Tmax < 10 min) or clearance (T1/2 < 10 min); normalized ERPF-OIH ≥ 100 mL/min or split ≥ 40%); pattern 1 (partial obstruction: Delayed peak (Tmax ≥ 10 min) or clearance (T1/2 ≥ 10 min) with a downward slope curve and normalized ERPF-OIH ≥ 100 mL/min or split ≥ 40%); pattern 2 (obstruction: Consistent upslope curve without a peak and normalized ERPF-OIH ≥ 100 mL/min or split ≥ 40%); pattern 3 (reduced uptake: 25%–40% split; and pattern 4 (severe reduced uptake: Split < 25%) (Appendix A).

In the control group, samples of voided bladder urine were collected once after enrollment. In the HN group, they were collected after enrollment; samples of RP urine were collected when patients underwent retrograde pyeloureterography with or without indwelling DJ stents (BARD INLAY® Ureteral Stent, Bard Inc, Murray Hill, NJ or Percuflex^TM^ Plus Stent, Boston Scientific Japan, Tokyo, Japan). In cases that underwent interventions for UUTO, such as DJ stent placement or surgery, bladder urine samples were collected 1 week and 4 weeks after the intervention. Four weeks after intervention, the status of UUTO was re-assessed with a second Tc-99mMAG3 renal scan (Appendix A). The median (interquartile range (IQR)) time from the initial diagnosis of HN to the first bladder urine collection in this study was 38 (16–269) and 199 (40–513) days in extrinsic and intrinsic UUTO, respectively. The median (IQR) time from the last assessment of HN before enrollment and the first MAG3 renal scan to the first bladder urine collection in all HN cases was 34 (14–92) and 8.5 (3–21) days, respectively, while the time from the first bladder urine collection to intervention was 1.5 (0–20) days. In patients who received surgery to improve their UUTO, DJ stents were not removed until the 4-week assessments were completed. The urine samples were stored at −80 °C until analysis.

Urinary vanin-1, KIM-1, and NGAL levels were measured using commercially available enzyme-linked immunosorbent assay (ELISA) kits (vanin-1, Cloud-Clone Corp, Houston, TX, USA; KIM-1 and NGAL, R&D Systems, Minneapolis, MN, USA), according to the manufacturer’s instructions. Urinary NAG levels were measured using an enzymatic colorimetric method (SRL, Tokyo, Japan), and values were normalized to urinary creatinine. Other clinical data, such as serum levels of creatinine and eGFR, were collected from electrical clinical records and we used the data within 1 week of the planned urine collection.

### Statistical Analyses

The D’Agnostino-Pearson omnibus normality test was used to test if the values followed a normal distribution. The test showed that all urinary AKI biomarkers and serum levels of creatinine were non-normally distributed while eGFR was normally distributed. The chi square test was used to analyze the relationship between two categorical variables. Comparisons between factors were made using the *t*-test (normal distribution) or the Mann–Whitney *U*-test (non-normal distribution). One-way analysis of variance (ANOVA) with Dunn’s multiple comparisons (normal distribution) or the Kruskal–Wallis test with Dunn’s multiple comparisons (non-normal distribution) was used to compare three or more variables. Two-way repeated-measures ANOVA with Tukey’s multiple comparisons test and two-way ANOVA with Sidak’s multiple comparisons test were used to analyze different categorical independent variables. Correlations between factors were analyzed using Pearson’s or Spearman’s correlation coefficient analysis. The cut-off values for each biomarker were determined based on the highest likelihood ratio in the ROC curve analyses. Logistic regression univariate and multivariate analyses were performed to determine independent predictive factor(s). All values are presented as mean ± standard deviation and *p* < 0.05 was considered significant. Statistical analyses were performed using GraphPad Prism (ver. 7.0; GraphPad, La Jolla, CA, USA) or SPSS for Windows software (ver. 19.0; SPSS Inc., Chicago, IL, USA).

## 5. Conclusions

This pilot study demonstrates that RP vanin-1 is highly sensitive and specific as a kidney injury biomarker for UUTO. It is also useful for assessing the severity of UUTO and monitoring the clinical course of UUTO. The diagnostic value of vanin-1 seems to be higher than that of KIM-1 or NGAL. Although bladder vanin-1 is less sensitive than RP vanin-1, the combined assessment of bladder vanin-1 and NAG could compensate for the low sensitivity of bladder vanin-1.

## Figures and Tables

**Figure 1 ijms-20-00899-f001:**
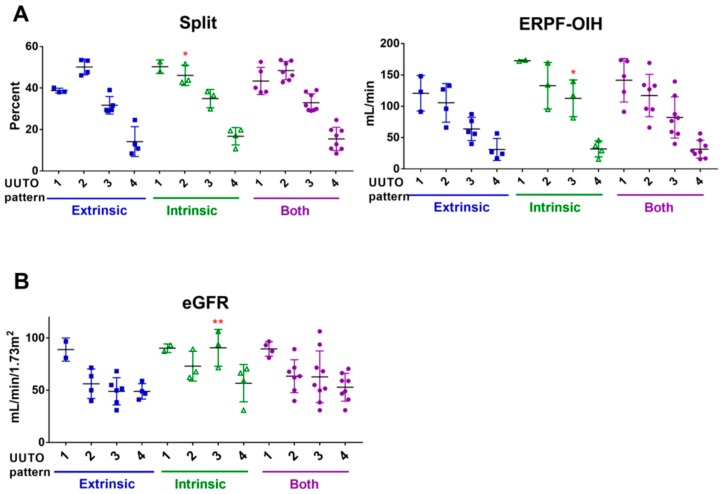
Renal functions according to UUTO patterns in the hydronephrosis group. The split and normalized ERPF-OIH in the affected kidney in the 99mTc-MAG3 renal scan (**A**) and eGFR (**B**) according to upper urinary tract obstruction (UUTO) patterns in extrinsic (blue squares), intrinsic (green triangles), and all UUTO (purple circles) cases. Generally, these parameters decreased with an increasing UUTO pattern. * *p* < 0.05, ** *p* < 0.01 vs. extrinsic UUTO by two-way analysis of variance (ANOVA) with Sidak’s multiple comparisons test.

**Figure 2 ijms-20-00899-f002:**
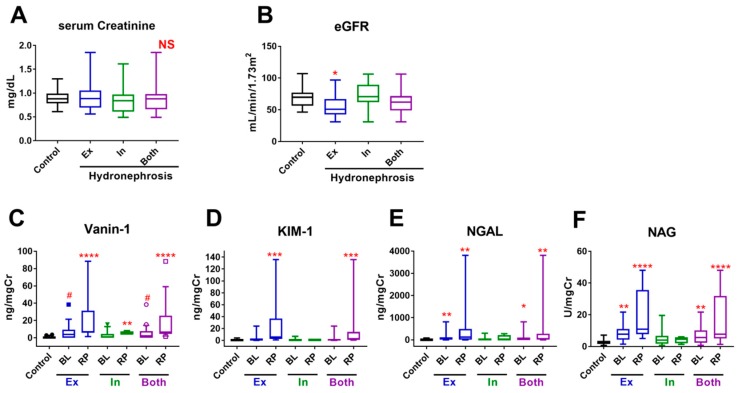
Renal functions and AKI biomarkers in the control versus hydronephrosis group. (**A**,**B**) The serum levels of creatinine (**A**) and eGFR (**B**) in the control and hydronephrosis groups (control, black; extrinsic UUTO, blue; intrinsic UUTO, green; all UUTO, purple). (**C**–**F**) The levels of vanin-1 (**C**), KIM-1 (**D**), NGAL (**E**), and NAG (**F**) in the bladder and renal pelvic urine in the hydronephrosis group (extrinsic UUTO, blue; intrinsic UUTO, green; and all UUTO, purple) were compared to the levels of each biomarker in the bladder urine of the control group (black). Data were analyzed using the Mann-Whitney *U* test (**A**), *t*-test (**B**), and the Kruskal-Wallis test with Dunn’s multiple comparisons test (**C**–**F**). # *p* < 0.1, * *p* < 0.05, ** *p* < 0.01, *** *p* < 0.01 **** *p* < 0.0001 vs. control. Ex, extrinsic UUTO; In, Intrinsic UUTO; NS, not significant.

**Figure 3 ijms-20-00899-f003:**
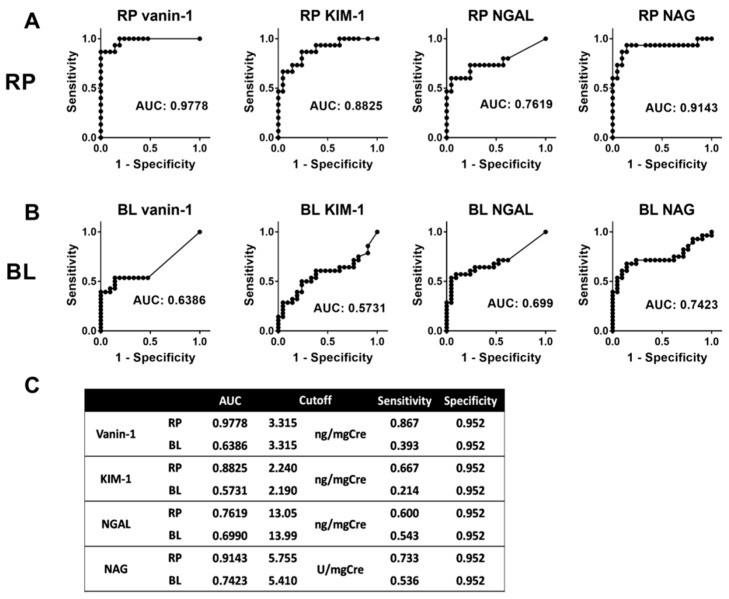
ROC curve analysis for each AKI biomarker. The area under the curve (AUC) of the receiver operating characteristics (ROC) curve analysis for each AKI biomarker in renal pelvic (**A**) and bladder urine (**B**) are shown. AUC, cutoff, sensitivity, and specificity in each AKI biomarker are shown in Table (**C**). RP, renal pelvis; BL, bladder; AUC, area under curve.

**Figure 4 ijms-20-00899-f004:**
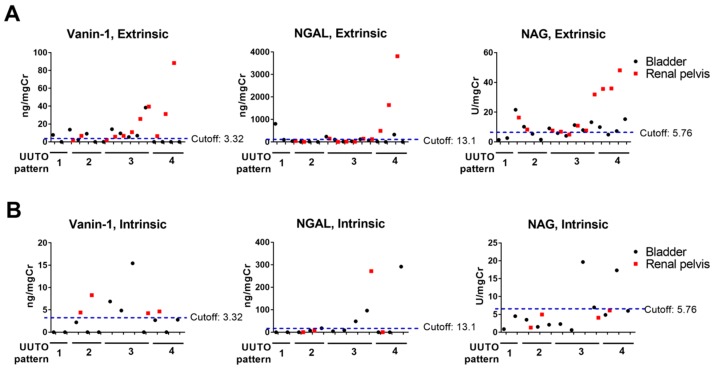
The levels of AKI biomarkers in the bladder and renal pelvic urine according to the UUTO pattern. The levels of vanin-1, NGAL, and NAG in the bladder and renal pelvic urine according to the UUTO pattern in extrinsic (**A**) and intrinsic UUTO (**B**) are shown. Black and red dots indicate the values in bladder urine and renal pelvic urine, respectively. Cut-off for each biomarker in RP urine (blue dashed line) was also shown.

**Figure 5 ijms-20-00899-f005:**
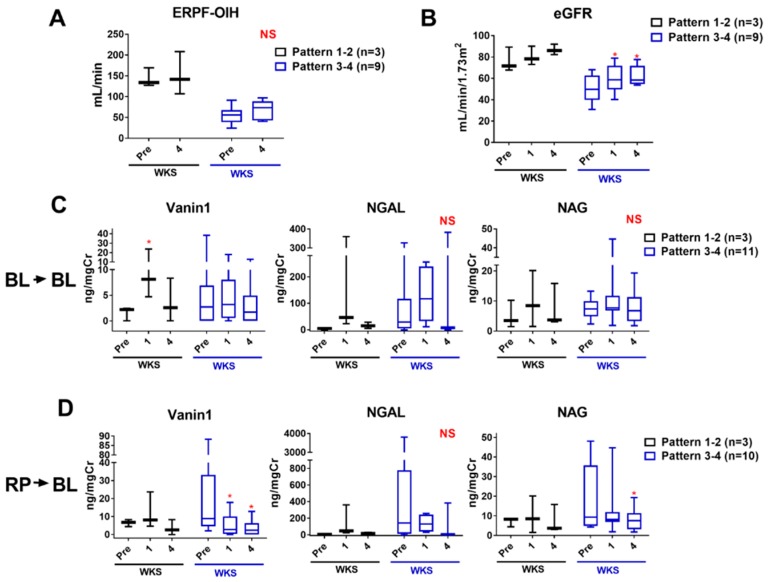
Changes in renal function and each AKI biomarker after intervention. (**A**,**B**) Changes in renal functions, as assessed by normalized ERPF-OIH (**A**) and eGFR (**B**), before and after intervention in pattern 1–2 (black) and pattern 3–4 UUTO (blue) cases are shown. (**C**,**D**) Changes in each AKI biomarker before and after intervention in pattern 1–2 (black) and pattern 3–4 UUTO (blue) cases are shown. The levels of each bladder (**C**) and renal pelvic biomarker (**D**) before intervention were compared to those of each bladder biomarker after intervention. Data were analyzed using two-way repeated measures ANOVA with Tukey’s multiple comparisons test. * *p* < 0.05 vs. pre (before intervention, pre); NS, not significant; wks, weeks; BL: bladder; RP renal pelvis.

**Figure 6 ijms-20-00899-f006:**
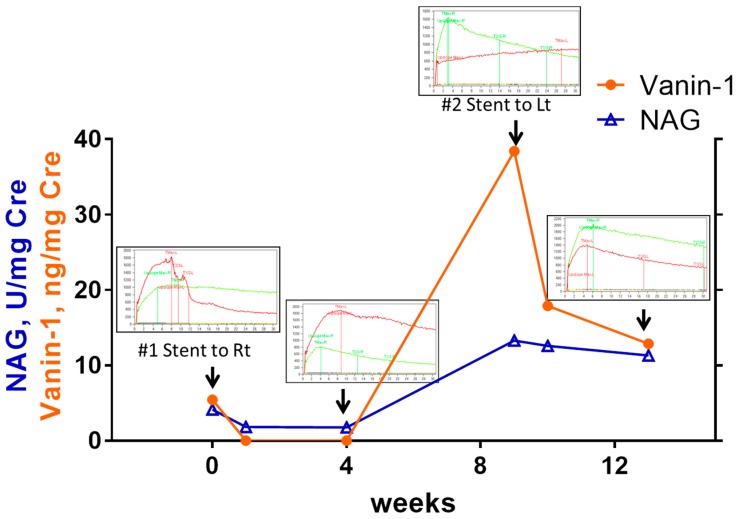
Changes in vanin-1 and NAG in a patient with asynchronous bilateral hydronephroses. Changes in vanin-1 (orange line and circles) and NAG (blue line and triangles) in a patient with asynchronous bilateral hydronephrosis are shown. Changes in MAG3 renal scan results are also shown. In each case, a ureteral double J stent was placed in the affected kidney.

**Table 1 ijms-20-00899-t001:** Characteristics of cases.

Charactesitics	Details	Control		HN					
		*n* = 21		Extrinsic, *n* = 16		Intrinsic, *n* = 12		Total, *n* = 28	
**Age, mean ± SD**		67.5 ± 10.73		58.63 ± 13.54		47.48 ± 13.87		53.85 ± 14.55 *	
**Sex, n (%)**	**Men**	19	(89)	6	(38)	6	(50)	12	(43)
	**Women**	2	(11)	10	(62)	6	(50)	16 *	(57)
**Affected kidney, n (%)**	**Right (Rt)**	NA		9	(56)	5	(42)	14	(50)
	**Left (Lt)**	NA		7	(44)	7	(58)	14	(50)
**Kidney disease other than HN, n (%)**	**Diabetic nephropathy**	3	(14)	0	(0)	0	(0)	0	(0)
	**Hypertensive nephropathy**	3	(14)	0	(0)	0	(0)	0	(0)
	**Drug-induced nephropathy**	0	(0)	1	(6)	1	(8)	2	(7)
	**Total**	6	(29)	1	(6)	1	(8)	2	(7)

* *p* < 0.001 vs. the control; HN, hydronephrosis; NA, not assessed.

**Table 2 ijms-20-00899-t002:** Univariate and multivariate analysis to identify the independent factors that predict the presence of hydronephrosis.

Variables		Univariate Analysis			Multivariate Analysis		
	N	HR	95% CI	*p* Value	HR	95% CI	*p* Value
**eGFR, mL/min/1.73 m^2^**	37	(-)	(-)	0.534	(-)	(-)	0.262
**Vanin-1, ng/mgCre**	37	1.445	1.042–2.008	0.028	1.546	1.046–2.283	0.029
**KIM-1, ng/mgCre**	37	(-)	(-)	0.319	(-)	(-)	0.224
**NGAL, ng/mgCre**	37	1.030	0.999–1.060	0.060	(-)	(-)	0.251
**NAG, U/mgCre**	37	1.447	1.092–1.919	0.010	1.458	1.067–1.992	0.018

Data could not be calculated and represented as hyphen (-). N, number of cases; HR, hazard ratio; CI, confidence interval.

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
