# Peer review of "A Novel Biomarker for Acute Kidney Injury, Vanin-1, for Obstructive Nephropathy: A Prospective Cohort Pilot Study"

_ijms, 2019, doi:10.3390/ijms20040899_

Reviewer 1 Report

In this manuscript Washino et al. investigate whether Vanin-1 is a novel biomarker of obstructive nephropathy. With a small sample size, results suggest that Vanin-1 in renal pelvic urine is a sensitive and specific biomarker for obstructive nephropathy. Overall the manuscript is well written, and provides some novel results.

Major concerns:

1.    Authors conclude that the Vanin-1 biomarker is a sensitive and specific AKI biomarker. Was the degree of AKI actually assessed in this study? and according to RIFLE/AKIN criteria?  Can the authors also provide a figure to illustrate the timing of diagnosis of HN/timing of the urine collection, Tc-99mMAG3 scan etc. 

2.    The results are interesting, but it remains unclear how this biomarker would be clinically useful. Levels of Vanin-1 were only detected in RP urine and not in bladder urine. This really limits the clinical utility of this biomarker. Imaging can already assess HN degree of UUTO. 

3.    Stats section is vague and does not fully replicate what was done in this study. Please expand this section. Are the biomarker levels normally distributed? This was not mentioned. 

4.    This is very limited information on the patient characteristics only Sex, AGE and eGFR is displayed. Other kidney disease could confound these biomarker levels. Do you have access to this information?

Minor comments:

5. Lines 45: another important clinical aspect is early detection of AKI. It is well known that creatinine lags behind Kidney injury. Include sentence about whether these markers are more sensitive than creatinine.

6. Abstract lines 21 I like the word ‘cases’ reserved for your population of interest i.e. HN. I find it a little confusing that you use cases to define both your control and HN groups. See line 64, you use cases for your HN group and not control. Please be consistent.

Line60: Same as above.

7. Line 96/ Figure 1: What is your rational for your choice of statistical test for multiple comparisons? Here you use Sidak’s and Figure 5 you use Tukey’s.

8. Figure 2: Should be using one-way ANOVA with multiple comparisons from control.

9. Table 2: Please display all stats for eGFR and KIM1 even if not significant.

10. Lines 142-144: please include AUC, Cut-off and sensitivity and specificity in a table.

11. Line 166: Can I see the ROC curve and the AUC for the combination of Vanin-1 + NGAL. In my opinion the sample size is too small for this analysis, but I am interested as an exploratory analysis.

12. Lines 237-239: Levels are below the cut-off in bladder because cut-offs were determined from RP urine.

13. Lines 239-242: For bladder Vanin-1, if urine flow is the issue from the effected kidney, how then do you explain that NAG was elevated in pattern 4 but not Vanin-1? You say that NAG might be absorbed from the affected kidney and excreted from contralateral kidney. Is there evidence of this? Or is the speculative? How does this compare with Vanin-1?

14. Lines 247-250: This greatly limits the clinical utility of Vanin-1. Can the authors comment on how this marker would be useful if useful measures cannot be non-inversely determined?

15. Lines 335-336: what was the timing of the measurement of serum creatinine and eGFR?

      Line 339: Was Chi-Squared test done for categorical variables i.e.Sex/ Affected kidney?

Author Response

Thank you so much for reviewing our manuscript and gave us an insightful comment. We revised our manuscript according to your comments and suggestion. Please check it out

1. UUTO includes various status of kidney injury, AKI, CRF, and ESRD, usually occurs in one upper urinary tract and, therefore, does not necessarily meet the criteria of AKI. We actually tested if urinary AKI biomarkers, including vanin-1, are useful for the evaluation of UUTO. Therefore, the conclusion of “the Vanin-1 biomarker is a sensitive and specific AKI biomarker” in the abstract section was inappropriate. We rephrased that (Line 31-32).

We added the figure to illustrate the timing of sample collection and renal scan in Fig. S4 (Line 411-418). Please check it out.

2. We think reviewer’s comment is partially true. Although it’s impossible to precisely assess the severity of UUTO unless RP vanin-1 levels are checked, most of bladder vanin-1 levels are above cutoff in extrinsic UUTO pattern 1-3 and in intrinsic UUTO pattern 3 (Fig 4). Additionally, the bladder vanin-1 levels seem to be increased according to the increase of UUTO pattern (Fig 4, Black dot).

We believe that, at least, bladder vanin-1 is useful for the initial assessment of HN, and cases with higher bladder vanin-1 levels have some kidney injury and would need further assessment. MAG3 renal scan is actually necessary at this point to distinguish cases very mild UUTO cases with complete UUTO cases unless RP vanin-1, more invasive examination, is assessed. However, only a few institutes can conduct MAG3 renal scan, which are expensive and expose patients to radiation. Therefore we need others biomarker to overcome this down side characteristics of vanin-1. We do believe combined assessment of serum or other bladder urinary biomarkers with bladder vanin-1 could distinguish cases with very mild UUTO with complete UUTO. At least, we showed the combination of bladder vanin-1 and NAG yielded better diagnostic values compared to each one.

3. We agreed with the reviewer’s comment. We re-analyzed the distribution of values, which demonstrated the distribution of these biomarkers was non-normal, so we re-analyzed the data of non-normal distribution using non-parametric way, which led to somewhat different results. In new statistical results, bladder vanin-1 levels in HN cases were marginally higher than control, and RP KIM-1 levels were significantly higher in HN cases than control. We changed the figure 2 and rephrased the results section accordingly (Line 25-26 and Line 106-116). We also added the results of ROC curve analysis and sensitivity/specificity of KIM-1 (Fig. 3, Line 143-155) because RP KIM-1 was significantly higher than control in new analyses. However, KIM-1 was still less useful for UUTO assessment than vanin-1 and we omitted the further assessment of KIM-1 from Fig. 4. We also rephrased the statistical analyses section in patients and methods. Please see Line 364-374.

4. We added information about concurrent kidney diseases other than HN in Table 1. We added some comments accordingly (Line 78-80). Please check it out.

Minor comments:

5. We added phrases accordingly. Please see Line 46.

6. We rephrased accordingly. Please see Line 21-22 and Line 64-65.

7. We used Tukey’s when we were comparing every row (or column) mean with every other row (or column) mean, and we used Sidak’s when we were comparing a bunch of independent comparisons independent comparisons according to the Graph Pad Prism recommendation.

8. The categories in Figure 2A and 2B were not independent, i.e. “cases with HN both equals to HN extrinsic + HN intrinsic”, so we believe statistical analysis for these data should be done using Student t-test or Mann-Whitney U test. In Figure 2C-F, we reanalyzed the data using Kruskal-Wallis test with Dunn’s multiple comparison test (similar to one-way ANOVA but non-parametric analysis) between Control vs. BL values vs. RP values in each extrinsic UUTO, intrinsic UUTO and both UUTO, respectively. These data were non-normally distributed and that’s why we used this statistical analysis. Re-analyses led to somewhat different results as commented in the “Response for major concern #3”.

 9. We did this analysis by logistic regression analysis with stepwise and likelihood methods using SPSS. In this analysis, stats for eGFR and KIM-1 do not show up. Therefore we cannot display these.

10. We added the table for AUC, Cutoff, Sensitivity and Specificity of each AKI biomarker as shown in Fig 3C. Please check it out (Line 155-156).

11.  ROC curve analysis for combination of two factors is very complicated and I could not figure out how to do it, but we got the sensitivity and specificity when vanin-1 (3.32 ng/mgCre) or NGAL (13.05 ng/mgCre) above cutoff was positive. In this analysis, sensitivity and specificity was 0.6552 and 0.9048, respectively, which was similar to combination of vanin-1 + NAG.

12. we determined cutoff from RP values because bladder vanin-1 and NGAL levels were less reliable. Most of these biomarker levels in pattern 4 UUTO were undetectable in spite of severe obstruction and sever kidney injury (higher RP values, Fig 4A). Furthermore, the levels of bladder and RP levels in each AKI biomarker was not quite different in pattern 1-3 UUTO (Fig 4A). We believe the low sensitivity of bladder vanin-1/KIM-1/NGAL was mainly due to undetectable or very low values in pattern 4 UUTO.

13. Our comment of “NAG might be absorbed from the affected kidney and excreted from contralateral kidney” is speculation. We checked previous papers about NAG, but there was no papers indicating this speculation. We added phrases of “although there has been no evidence to date that demonstrate our speculation” (Line 297). We need to assess if this speculation is correct using complete UUTO model rats and we are now considering it seriously.

14. we added comments about this issue. Please see Line 260-268.

15. we added phrases about this. Please see Line 362.

Reviewer 2 Report

Dear Authors,

I read your article and it seems to me exceptional.

1. I recommend introducing Study Flowchart for easier tracking of the study.

2. I also recommend the introduction of some sections (introduction / discussion) about microRNAs - another interesting biomarker. maybe even compare in 2-3 phrases microRNAs with what you have studied (Please see some references: https://www.ncbi.nlm.nih.gov/pubmed/30551680 , https://www.ncbi.nlm.nih.gov/pubmed/29739062 , https://www.ncbi.nlm.nih.gov/pubmed/28164521 )

Otherwise, small writing mistakes that you will please correct.

Author Response

Thank you so much for reviewing our manuscript and gave us an insightful comment. We revised our manuscript according to your comments and suggestion. Please check it out.

Point by point response

1. We added Flowchart in Fig S4. Please check it out (Line 411-418).

2. We agree with this comment. miRNA seems to be promising and serum levels of these biomarkers are useful for detecting AKI. These biomarkers might be useful for UUTO. We added phrases about miRNA. Please see Line 301-307.

Round  2

Reviewer 1 Report

I would again like to congratulate the authors on their work. There are a few outstanding issues I would like addressed regarding stats and conclusions.

1.     Fig S4. It is still difficult to discern the timing of urine collection. I am more interested in the timing from Diagnosis of HN to enrollment and from enrollment to intervention. Do you have the median time from diagnosis of HN to enrollment. Report this in methodology.

2.     You used the highest likelihood ratio in ROC curve analysis to derive your cut-offs. In one of your responses you say that cut-offs are from RP urine. This needs to be specified in methods/results. In Figure 3C Table, the cut-offs should correspond with RP urine not both.

3.     Re: Major concern #2:

We think reviewer’s comment is partially true. Although it’s impossible to precisely assess the severity of UUTO unless RP vanin-1 levels are checked, most of bladder vanin-1 levels are above cutoff in extrinsic UUTO pattern 1-3 and in intrinsic UUTO pattern 3 (Fig 4). Additionally, the bladder vanin-1 levels seem to be increased according to the increase of UUTO pattern (Fig 4, Black dot).

The results that the authors present on this novel biomarker are interesting but I believe that the conclusions from the BL urine are still inflated. With the new non-parametric analysis, levels of bladder Vanin-1 are not significantly different than controls. From 3B The ROC curve AUC is only 0.63. Looking at Figure 4, BL urine vanin-1 levels from patterns 1-3 UUTO 4/12 are below the cut-off. The authors should go through the discussion and tone down conclusions from bladder urine. The results from RP urine are interesting and should receive the focus.

4.     Lines 129-136 + Table 2: If biomarkers are non-parametric, log-transformed values should be used in the logistical regression. The logistical regression is a parametric test. It wouldn’t make sense that BL levels are not different than controls, but significant in the univariate analysis? This analysis should be re-done using log-transformed values.

5.     Conclusion Lines:380: “RP urine vanin-1 is highly sensitive”. No evidence that BL urine is sensitive. 

Author Response

Thank you so much for reviewing our manuscript again. We revised our manuscript as much as we can according to the reviewer’s comment. Please check it out.

1.  I added those data in the method sections. Please see Line 351-356. We used the timing of the first bladder urine collection, instead of time of enrollment because it represents the actual time to collect sample. We assessed time from initial diagnosis of HN to the initial bladder urine collection and time from the last assessment of HN before enrollment to the initial bladder urine collection. In some cases, there were long time lags from the initial diagnosis to enrollment because some of mild obstructive cases were not referred to our department at the first diagnosis of HN and referred once it became worse. We also assessed time from MAG3 renal scan to the initial bladder urine collection, and that from initial bladder urine collection to the intervention. Please check it out.

2.  We agree with this comment. We re-analyzed cutoffs of each marker in RP and BL urine, respectively, using the highest likelihood ratio in ROC curve analysis of RP and BL urine. The cutoffs between RP and BL urine were quite similar but somewhat different in KIM-1, NGAL, and NAG. We changed the sensitivity and specificity in BL urine accordingly (Fig 3C and Line 147-152). In the combined assessment of BL vanin-1 and NAG, sensitivity and specificity were still 0.6786 and 0.9048 in the new cutoff setting (Results section 2.7.).

3.  We agree with BL vanin-1 in the initial assessment was less useful than RP vanin-1. We omitted some comments about BL vanin-1 in the Discussion section (Line 259-265) and changed the conclusion (Line 381) according to reviewer’s suggestion. Please check it out.

4.  Our data have lots of undetectable variables, especially in vanin-1, which is expressed as “0”. We can’t log-transform that “0”.  Additionally, in my understanding, there is no assumption about normality on independent variable in multiple logistic regression analysis and the fit does not require normality (https://www.researchgate.net/post/Should_I_transform_non-normal_independent_variables_in_logistic_regression).                                                                                        

5.  We agree with this comment and changed the sentence accordingly. Please see Line 384.

Best regards,

Satoshi Washino